# Conditional Generative Models are Provably Robust: Pointwise Guarantees for Bayesian Inverse Problems

**Fabian Altekrüger**                                        *fabian.altekrueger@hu-berlin.de*
*Department of Mathematics*
*Humboldt-Universität zu Berlin*

**Paul Hagemann**                                            *hagemann@math.tu-berlin.de*
*Institute of Mathematics*
*Technische Universität Berlin*

**Gabriele Steidl**                                          *steidl@math.tu-berlin.de*
*Institute of Mathematics*
*Technische Universität Berlin*

**Reviewed on OpenReview:** *https://openreview.net/forum?id=Wcui061fxr*

## Abstract

Conditional generative models became a very powerful tool to sample from Bayesian inverse problem posteriors. It is well-known in classical Bayesian literature that posterior measures are quite robust with respect to perturbations of both the prior measure and the negative log-likelihood, which includes perturbations of the observations. However, to the best of our knowledge, the robustness of conditional generative models with respect to perturbations of the observations has not been investigated yet. In this paper, we prove for the first time that appropriately learned conditional generative models provide robust results for single observations.

## 1 Introduction

Recently, the use of neural networks (NN) in the field of uncertainty quantification has emerged, as one often is interested in the statistics of a solution and not just in point estimates. In particular, Bayesian approaches in inverse problems received great interest. In this paper, we are interested in learning the whole posterior distribution in Bayesian inverse problems by conditional generative NNs as proposed, e.g., in (Adler & Öktem, 2018; Ardizzone et al., 2019; Batzolis et al., 2021; Hagemann et al., 2022). Addressing the posterior measure instead of end-to-end reconstructions has several advantages as illustrated in Figure 1. More precisely, if we consider a Gaussian mixture model as prior distribution and a linear forward operator with additive Gaussian noise, then the posterior density (red) can be computed explicitly. Obviously, these curves change smoothly with respect to the observation $y$, i.e., we observe a continuous behaviour of the posterior also with respect to observations near zero. In particular, (samples of) the posterior can be used to provide

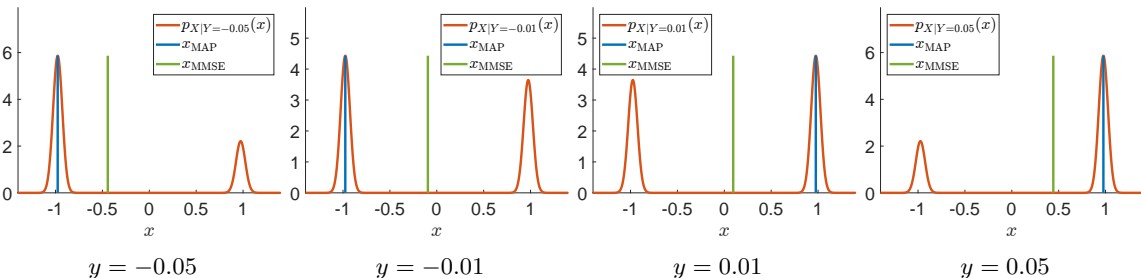

Figure 1: Posterior density (red), MAP estimator (blue) and MMSE estimator (green) for different observations $y = -0.05, -0.01, 0.01, 0.05$ (from left to right). While the MAP estimator is discontinuous with respect to the observation $y$, the posterior density is continuous with respect to y. The MMSE estimator just gives the expectation value of the posterior which is, in contrast to MAP, not the value with highest probability.

additional information on the reconstructed data, for example on their uncertainty. As can be seen in the figure, for fixed $y$ the minimum mean squared error (MMSE) estimator delivers just an averaged value. In contrast to the non robust maximum a-posteriori (MAP) estimator, this is not the one with the highest probability. Even worse, the MMSE can output values which are not even in the prior distribution. For a more comprehensive comparison of MAP estimator, MMSE estimator and posterior distribution, we refer to Appendix A

Several robustness guarantees on the posterior were proved in the literature. One of the first results in the direction of stability with respect to the distance of observations was obtained by Stuart (2010) with respect to the Hellinger distance, see also Dashti & Stuart (2017). A very related question instead of perturbed observations concerns the approximations of forward maps, which was investigated by Marzouk & Xiu (2009). Furthermore, different prior measures were considered in (Hosseini, 2017; Hosseini & Nigam, 2017; Sullivan, 2017), where they also discuss the general case in Banach spaces. Two recent works (Latz, 2020; Sprungk, 2020) investigated the (Lipschitz) continuity of the posterior measures with respect to a multitude of metrics, where Latz (2020) focused on the well-posedness of the Bayesian inverse problem and Sprungk (2020) on the local Lipschitz continuity. Most recently, in Garbuno-Inigo et al. (2023) the stability estimates have been generalized to integral probability metrics circumventing some Lipschitz conditions done in Sprungk (2020). Our paper is based on the findings of Sprungk (2020), but relates them with conditional generative NNs that aim to learn the posterior.

More precisely, in many machine learning papers, the following idea is pursued in order to solve inverse problems simultaneously for all observations $y$: Consider a family of generative models $G_\theta(y, \cdot)$ with parameters $\theta$, which are supposed to map a latent distribution, like the standard Gaussian one, to the absolutely continuous posteriors $P_{X|Y=y}$, i.e., $G_\theta(y, \cdot)_\# P_Z \approx P_{X|Y=y}$. In order to learn such a conditional generative model, usually a loss of the form

$$L(\theta) := \mathbb{E}_{y \sim P_Y}[D(P_{X|Y=y}, G_\theta(y, \cdot)_\# P_Z)]$$

is chosen with some "distance" $D$ between measures like the Kullback-Leibler (KL) divergence $D = \text{KL}$ used in Ardizzone et al. (2019) for training normalizing flows or the Wasserstein-1 distance $D = W_1$ appearing, e.g., in the framework of (conditional) Wasserstein generative adversarial

networks (GANs) (Adler & Öktem, 2018; Arjovsky et al., 2017; Liu et al., 2021). Also conditional diffusion models (Igashov et al., 2022; Song et al., 2021b; Tashiro et al., 2021) fit into this framework. Here De Bortoli (2022) showed that the standard score matching diffusion loss also optimizes the Wasserstein distance between the target and predicted distribution.

However, in practice we are usually interested in the reconstruction quality from a single or just a few measurements which are null sets with respect to $P_Y$. In this paper, we are interested in the important question, whether there exist any guarantees for the NN output to be close to the posterior for one specific measurement $\tilde{y}$. Our main result in Theorem 5 shows that for a NN learned such that the loss becomes small in the Wasserstein-1 distance, say $L(\theta) < \varepsilon$, the distance $W_1(P_{X|Y=\tilde{y}}, G_\theta(\tilde{y}, \cdot)_\# P_Z)$ becomes also small for the single observation $\tilde{y}$. More precisely, we get the bound

$$W_1(P_{X|Y=\tilde{y}}, G_\theta(\tilde{y}, \cdot)_\# P_Z) \leq C \varepsilon^{\frac{1}{n+1}},$$

where $C$ is a constant and $n$ is the dimension of the observations. To the best of our knowledge, this is the first estimate given in this direction.

We like to mention that in contrast to our paper, where we assume that samples are taken from the distribution for which the NN was learned, Hong et al. (2022) observed that conditional normalizing flows are unstable when feeding them out-of-distribution observations. This is not too surprising given some literature on the instability of (conditional) normalizing flows (Behrmann et al., 2021; Kirichenko et al., 2020).

**Outline of the paper.** The main theorem is shown in Section 2. For this we introduce several lemmata for the local Lipschitz continuity of posterior measures and conditional generative models with respect to the Wasserstein distance. In Section 3, we discuss the dependence of our derived bound on the training loss for different conditional generative models. In Appendix A, we illustrate by a simple example with a Gaussian mixture prior and Gaussian noise, why posterior distributions can be expected to be more stable than maximum a-posteriori (MAP) estimations and have more desirable properties than minimum mean squared error (MMSE) estimations.

## 2 Pointwise Robustness of Conditional Generative NNs

Let $X \in \mathbb{R}^m$ be a continuous random variable with law $P_X$ determined by its density function $p_X$ and $f \colon \mathbb{R}^m \to \mathbb{R}^n$ a measurable function. We consider a Bayesian inverse problem

$$Y = \mathrm{noisy}(f(X)) \tag{1}$$

where "noisy" describes the underlying noise model. A typical choice is additive Gaussian noise, resulting in

$$Y = f(X) + \Xi, \quad \Xi \sim \mathcal{N}(0, \sigma^2 I_n).$$

Let $G_\theta = G \colon \mathbb{R}^n \times \mathbb{R}^d \to \mathbb{R}^m$ be a conditional generative model trained to approximate the posterior distribution $P_{X|Y=y}$ using the latent random variable $Z \in \mathbb{R}^d$. We will assume that all appearing measures are absolutely continuous and that the first moment of $G(y, \cdot)_\# P_Z$ is finite for all $y \in \mathbb{R}^n$. In particular, the posterior density is related via Bayes' theorem through the prior $p_X$ and the likelihood $p_{Y|X=x}$ as

$$p_{X|Y=y} \propto p_{Y|X=x} p_X,$$

where $\propto$ means quality up to a multiplicative normalization constant. Since the posterior density $p_{X|Y=y}$ is only almost everywhere unique, we choose the unique continuous representative. Further, we assume that the negative log-likelihood $-\log p_{Y|X=x}$ is bounded from below with respect to $x$, i.e., $\inf_x -\log p_{Y|X=x} > -\infty$. In particular, this includes mixtures of additive and multiplicative noise $Y = f(X) + \Xi_1 + \Xi_2 f(X)$, if $X$, $\Xi_1$ and $\Xi_2$ are independent, or log-Poisson noise commonly arising in computerized tomography.

We will use the Wasserstein-1 distance (Villani, 2009), which is a metric on the space of probability measures with finite first moment and is defined for measures $\mu$ and $\nu$ on the space $\mathbb{R}^m$ as

$$W_1(\mu, \nu) = \inf_{\pi \in \Pi(\mu, \nu)} \int_{\mathbb{R}^m \times \mathbb{R}^m} \|x - y\| d\pi(x, y),$$

where $\Pi(\mu, \nu)$ contains all measures on $\mathbb{R}^m \times \mathbb{R}^m$ with $\mu$ and $\nu$ as its marginals. The Wasserstein distance can be also rewritten by its dual formulation (Villani, 2009, Remark 6.5) as

$$W_1(\mu, \nu) = \max_{\mathrm{Lip}(\varphi) \leq 1} \int \varphi(x) \mathrm{d}(\mu - \nu)(x). \tag{2}$$

First, we show the local Lipschitz continuity of our generating measures $G(y, \cdot)_{\#} P_Z$ with respect to $y$, where we assume a local boundedness of the Jacobian of the generator with respect to the observation.

**Lemma 1** (Local Lipschitz continuity of generator). *For all $r > 0$, there exists some $L_r > 0$ such that for any parameterized family of generative models $G$ with $\|\nabla_y G(y, z)\| \leq L_r$ for all $z \in \mathrm{supp}(P_Z)$ and all $y \in \mathbb{R}^n$ with $\|y\| \leq r$ it holds*

$$W_1(G(y_1, \cdot)_{\#} P_Z, G(y_2, \cdot)_{\#} P_Z) \leq L_r \|y_1 - y_2\|$$

*for all $y_1, y_2 \in \mathbb{R}^n$ with $\|y_1\|, \|y_2\| \leq r$.*

*Proof.* We use the mean value theorem which yields for every $z \in \mathrm{supp}(P_Z)$ and all $y_1, y_2 \in \mathbb{R}^n$ with $\|y_1\|, \|y_2\| \leq r$

$$\|G(y_1, z) - G(y_2, z)\| = \left\| \int_0^1 \nabla_y G(y_1 + t(y_2 - y_1), z)(y_1 - y_2) \mathrm{d}t \right\|$$

$$\leq \int_0^1 \|\nabla_y G(y_1 + t(y_2 - y_1), z)\| \mathrm{d}t \|y_1 - y_2\|$$

$$\leq L_r \|y_1 - y_2\|.$$

Next, we apply the dual formulation of the Wasserstein-1 distance to estimate

$$W_1(G(y_1, \cdot)_{\#} P_Z, G(y_2, \cdot)_{\#} P_Z) = \max_{\mathrm{Lip}(\varphi) \leq 1} \mathbb{E}_{z \sim P_Z}[\varphi(G(y_1, z)) - \varphi(G(y_2, z))]$$

$$\leq \max_{\mathrm{Lip}(\varphi) \leq 1} \mathbb{E}_{z \sim P_Z}[|\varphi(G(y_1, z)) - \varphi(G(y_2, z))|]$$

$$\leq \mathbb{E}_{z \sim P_Z}[\|G(y_1, z) - G(y_2, z)\|]$$

$$\leq L_r \|y_1 - y_2\|.$$

$\square$

We want to highlight that there is a trade-off between regularity of the generator due to a small Lipschitz constant (Gouk et al., 2021; Miyato et al., 2018) and expressivity of the generator requiring a large Lipschitz constant (Hagemann & Neumayer, 2021; Salmona et al., 2022).

**Remark 2.** *If $P_Z$ is supported on a compact set, then the assumption in Lemma 1 is fulfilled for generators which are, e.g., continuously differentiable and then it follows from the extreme value theorem. Note that if we choose $P_Z$ to be a Gaussian distribution, then it holds $\operatorname{supp}(P_Z) = \mathbb{R}^d$. Thus, for continuously differentiable generators it is not clear that this assumption is fulfilled, but at least the weaker assumption $\|\nabla_y G(y, z)\| \le L_r$ for all $z \in \mathbb{R}^d$ with $\|z\| \le \tilde{r}$ and all $y \in \mathbb{R}^n$ with $\|y\| \le r$ holds true. In this case, we can show that Lemma 1 holds true up to an arbitrary small additive constant, see Appendix B for more details.*

By the following lemma, which is just (Sprungk, 2020, Corollary 19) for Euclidean spaces, the local Lipschitz continuity of the posterior distribution with respect to the Wasserstein-1 distance is guaranteed under the assumption of a locally Lipschitz likelihood.

**Lemma 3** (Local Lipschitz continuity of the posterior). *Let the forward operator $f$ and the likelihood $p_{Y|X=x}$ in (1) be measurable. Assume that there exists a function $M \colon [0, \infty) \times \mathbb{R} \to [0, \infty)$ which is monotone in the first component and non-decreasing in the second component such that for all $y_1, y_2 \in \mathbb{R}^n$ with $\|y_1\|, \|y_2\| \le r$ for $r > 0$ and for all $x \in \mathbb{R}^m$ it holds*

$$|\log p_{Y|X=x}(y_2) - \log p_{Y|X=x}(y_1)| \le M(r, \|x\|)\|y_1 - y_2\|. \tag{3}$$

*Furthermore, assume that $M(r, \|\cdot\|) \in L^2_{P_X}(\mathbb{R}^m, \mathbb{R})$. Then, for any $r > 0$ there exists a constant $C_r < \infty$ such that for all $y_1, y_2 \in \mathbb{R}^n$ with $\|y_1\|, \|y_2\| \le r$ we have*

$$W_1(P_{X|Y=y_1}, P_{X|Y=y_2}) \le C_r \|y_1 - y_2\|.$$

The Lipschitz constants of the family of generative models and the posterior distributions $P_{X|Y=y}$ can be related to each other under some convergence assumptions. Let the assumptions of Lemma 3 be fulfilled, assume further that there exists a family of generative models $(G^\varepsilon)_\varepsilon$ satisfying

$$\lim_{\varepsilon \to 0} G^\varepsilon(y, \cdot)_\# P_Z = P_{X|Y=y}$$

with respect to the $W_1$-distance and consider observations $y_1, y_2 \in \mathbb{R}^n$ with $\|y_1\|, \|y_2\| \le r$. Then, by the triangle inequality it holds

$$\begin{aligned}
\lim_{\varepsilon \to 0} W_1(G^\varepsilon(y_1, \cdot)_\# P_Z, G^\varepsilon(y_2, \cdot)_\# P_Z) &\le \lim_{\varepsilon \to 0} W_1(G^\varepsilon(y_1, \cdot)_\# P_Z, P_{X|Y=y_1}) + W_1(P_{X|Y=y_1}, P_{X|Y=y_2}) \\
&\quad + W_1(P_{X|Y=y_2}, G^\varepsilon(y_2, \cdot)_\# P_Z) \\
&= W_1(P_{X|Y=y_1}, P_{X|Y=y_2}) \\
&\le C_r \|y_1 - y_2\|.
\end{aligned}$$

Hence, under the assumption of convergence, we expect the Lipschitz constant of our conditional generative models to behave similar to the one of the posterior distribution.

**Remark 4.** *The assumption (3) is for instance fulfilled for additive Gaussian noise $\Xi \sim \mathcal{N}(0, \sigma^2 \mathrm{Id})$. In this case*

$$-\log p_{Y|X=x}(y) = \frac{n}{2}\log(2\pi\sigma^2) + \frac{1}{2\sigma^2}\|y - f(x)\|^2.$$

*Hence* $-\log p_{Y|X=x}(y)$ *is differentiable with respect to* $y$ *and we get local Lipschitz continuity of the negative log-likelihood.*

Now we can prove our main theorem which ensures pointwise bounds on the Wasserstein distance between posterior and generated measure, if the expectation over $P_Y$ is small. In particular, the bound depends on the local Lipschitz constant of the conditional generator with respect to the observation which may depend on the architecture of the generative model, the local Lipschitz constant of the inverse problem, the expectation over $P_Y$ and the probability of the considered observation $\tilde{y}$. We want to highlight that the bound depends on the evidence $p_Y(\tilde{y})$ of an observation $\tilde{y}$ and indicates that we generally cannot expect a good pointwise estimate for out-of-distribution observations, i.e., $p_Y(y) \approx 0$. This is in agreement with the empirical results presented in Hong et al. (2022). The proof of Theorem 5 has a nice geometric interpretation, which is visualized in Figure 2.

**Theorem 5.** *Let the forward operator* $f$ *and the likelihood* $p_{Y|X=x}$ *in* (1) *fulfill the assumptions of Lemma 3. Let* $\tilde{y} \in \mathbb{R}^n$ *be an observation with* $p_Y(\tilde{y}) = a > 0$. *Further, assume that* $y \mapsto p_Y(y)$ *is differentiable with* $\|\nabla p_Y(y)\| \leq K$ *for* $K > 0$ *and all* $y \in \mathbb{R}^n$. *For fixed* $k = \frac{a}{2K} + \|\tilde{y}\|$, *assume that we have trained a family of generative models* $G$ *which fulfills* $\|\nabla_y G(y, z)\| \leq L_k$ *for all* $z \in \mathrm{supp}(P_Z)$ *and all* $y \in \mathbb{R}^n$ *with* $\|y\| \leq k$ *for some* $L_k > 0$ *such that*

$$\mathbb{E}_{y \sim P_Y}[W_1(P_{X|Y=y}, G(y, \cdot)_\# P_Z)] \leq \varepsilon \tag{4}$$

*for some* $\varepsilon > 0$. *Then we have for* $\varepsilon \leq 1$ *that*

$$W_1(P_{X|Y=\tilde{y}}, G(\tilde{y}, \cdot)_\# P_Z) \leq (L_k + C_k)\frac{\varepsilon^{\frac{1}{n+1}} a}{2K} + \frac{2\varepsilon^{\frac{1}{n+1}}}{S_n(\frac{a}{2K})^n a}, \tag{5}$$

*where* $S_n := \pi^{\frac{n}{2}}/\Gamma(\frac{n}{2}+1)$ *and* $C_\bullet$ *is the Lipschitz constant from Lemma 3. If* $\varepsilon$ *additionally satisfies* $\varepsilon \leq \left(\frac{a}{2K}\right)^{n+1} \frac{(L_k+C_k)S_n a}{2n}$, *then it holds*

$$W_1(P_{X|Y=\tilde{y}}, G(\tilde{y}, \cdot)_\# P_Z) \leq (L_k + C_k)^{1-\frac{1}{n+1}}\left(1 + \frac{1}{n}\right)\left(\frac{2n}{S_n a}\right)^{\frac{1}{n+1}} \varepsilon^{\frac{1}{n+1}}. \tag{6}$$

*Proof.* Let $0 < r \leq \frac{a}{2K}$. Then, for $y \in B_r(\tilde{y})$, there exists by the mean value theorem some $\xi \in \overline{y\tilde{y}}$ such that

$$|p_Y(y) - p_Y(\tilde{y})| \leq \|\nabla p_Y(\xi)\|\|y - \tilde{y}\| \leq Kr \leq \frac{a}{2}.$$

Consequently, each $y \in B_r(\tilde{y})$ has at least probability $p_Y(y) \geq \frac{a}{2}$. Moreover, by the volume of the $n$-dimensional ball it holds that

$$P_Y(B_r(\tilde{y})) = \int_{B_r(\tilde{y})} p_Y(y)\mathrm{d}y \geq \frac{\pi^{\frac{n}{2}}}{\Gamma(\frac{n}{2}+1)} r^n \frac{a}{2} = S_n r^n \frac{a}{2}.$$

Now we claim that there exists $\hat{y} \in B_r(\tilde{y})$ with

$$W_1(P_{X|Y=\hat{y}}, G(\hat{y}, \cdot)_\# P_Z) \leq \frac{2\varepsilon}{S_n r^n a}. \tag{7}$$

If this would not be the case, this would imply a contradiction to (4) by

$$
\begin{aligned}
\mathbb{E}_{y \sim P_Y}[W_1(P_{X|Y=y}, G(y, \cdot)_{\#}P_Z)] &= \int_{\mathbb{R}^n} W_1(P_{X|Y=y}, G(y, \cdot)_{\#}P_Z)\mathrm{d}P_Y(y) \\
&\geq \int_{B_r(\tilde{y})} W_1(P_{X|Y=y}, G(y, \cdot)_{\#}P_Z)\mathrm{d}P_Y(y) \\
&> \int_{B_r(\tilde{y})} \frac{2\varepsilon}{S_n r^n a}\mathrm{d}P_Y(y) \\
&= P_Y(B_r(\tilde{y}))\frac{2\varepsilon}{S_n r^n a} \geq \varepsilon.
\end{aligned}
$$

Next, we show the local Lipschitz continuity of $y \mapsto W_1(P_{X|Y=y}, G(y, \cdot)_{\#}P_Z)$ on $B_r(\tilde{y})$ by combining Lemma 1 and Lemma 3. Let $y_1, y_2 \in B_r(\tilde{y})$, so that $\|y_1\|, \|y_2\| \leq \|\tilde{y}\| + r$. Let $L_{\|\tilde{y}\|+r} > 0$ be the local Lipschitz constant from Lemma 1 and $C_{\|\tilde{y}\|+r}$ the local Lipschitz constant from Lemma 3. Using the triangle inequality and its reverse, we get

$$
\begin{aligned}
&|W_1(P_{X|Y=y_1}, G(y_1, \cdot)_{\#}P_Z) - W_1(P_{X|Y=y_2}, G(y_2, \cdot)_{\#}P_Z)| \\
&\leq |W_1(P_{X|Y=y_1}, G(y_1, \cdot)_{\#}P_Z) - W_1(P_{X|Y=y_1}, G(y_2, \cdot)_{\#}P_Z)| \\
&\quad + |W_1(P_{X|Y=y_1}, G(y_2, \cdot)_{\#}P_Z) - W_1(P_{X|Y=y_2}, G(y_2, \cdot)_{\#}P_Z)| \quad\quad (8) \\
&\leq W_1(G(y_1, \cdot)_{\#}P_Z, G(y_2, \cdot)_{\#}P_Z) + W_1(P_{X|Y=y_1}, P_{X|Y=y_2}) \\
&\leq (L_{\|\tilde{y}\|+r} + C_{\|\tilde{y}\|+r})\|y_1 - y_2\|.
\end{aligned}
$$

Combination of the results in (7) and (8) yields the estimate

$$
\begin{aligned}
W_1(P_{X|Y=\tilde{y}}, G(\tilde{y}, \cdot)_{\#}P_Z) &\leq |W_1(P_{X|Y=\tilde{y}}, G(\tilde{y}, \cdot)_{\#}P_Z) - W_1(P_{X|Y=\widehat{y}}, G(\widehat{y}, \cdot)_{\#}P_Z)| \\
&\quad + |W_1(P_{X|Y=\widehat{y}}, G(\widehat{y}, \cdot)_{\#}P_Z)| \\
&\leq (L_{\|\tilde{y}\|+r} + C_{\|\tilde{y}\|+r})r + \frac{2\varepsilon}{S_n r^n a} \\
&\leq (L_{\|\tilde{y}\|+\frac{a}{2K}} + C_{\|\tilde{y}\|+\frac{a}{2K}})r + \frac{2\varepsilon}{S_n r^n a}.
\end{aligned}
$$

If $\varepsilon \leq 1$, we can choose $r = \varepsilon^{\frac{1}{n+1}}\frac{a}{2K} \leq \frac{a}{2K}$ which results in (5). On the other hand, the radius $r$, for which the right-hand side becomes minimal, is given by

$$
r = \left(\frac{2n\varepsilon}{(L_{\|\tilde{y}\|+\frac{a}{2K}} + C_{\|\tilde{y}\|+\frac{a}{2K}})S_n a}\right)^{\frac{1}{n+1}}.
$$

Plugging this in, we get (6), which has the same asymptotic rate.. However, we need that $r \leq \frac{a}{2K}$ which implies

$$
\varepsilon \leq \left(\frac{a}{2K}\right)^{n+1} \frac{(L_{\|\tilde{y}\|+\frac{a}{2K}} + C_{\|\tilde{y}\|+\frac{a}{2K}})S_n a}{2n}.
$$

$\square$

Note that in (4) we assume that the expectation over $P_Y$ of the Wasserstein distance is small. When training a generator, usually a finite training set is available. The measure $P_Y$ can be approximated by the empirical training set with a rate $n^{-1/d}$ on compact sets, where $n$ is the size of the training set and $d$ the dimension, see, e.g., Weed & Bach (2019).

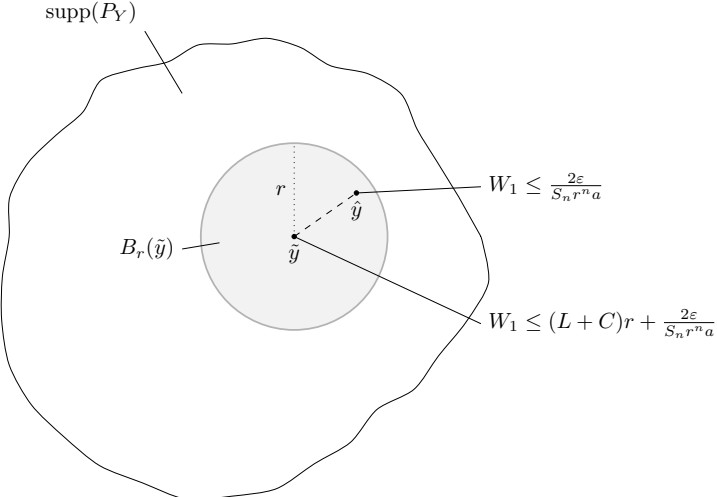

Figure 2: Geometric interpretation of the proof of Theorem 5. Inside the ball $B_r(\tilde{y})$ we can find some $\hat{y}$, for which the Wasserstein distance $W_1(P_{X|Y=\hat{y}}, G(\hat{y}, \cdot)_\# P_Z)$ is bounded by $\frac{2\varepsilon}{S_n r^n a}$. Using the regularity of the generator and of the inverse problem, the Wasserstein distance $W_1(P_{X|Y=\tilde{y}}, G(\tilde{y}, \cdot)_\# P_Z)$ can be bounded by the triangle inequality.

**Remark 6.** *We can get rid of the dimension scaling $\varepsilon^{\frac{1}{n+1}}$ by choosing the radius as $r = \frac{a}{2K}$, which yields*

$$W_1(P_{X|Y=\tilde{y}}, G(\tilde{y}, \cdot)_\# P_Z) \leq (L_{\|\tilde{y}\|+\frac{a}{2K}} + C_{\|\tilde{y}\|+\frac{a}{2K}})\frac{a}{2K} + \frac{2\varepsilon}{S_n(\frac{a}{2K})^n a}.$$

*This comes at the disadvantage that the first term is constant with respect to $\varepsilon$.*

The following corollary provides a characterization of a perfect generative model. If the expectation (4) goes to zero, then for all $y \in \mathbb{R}^n$ with $p_Y(y) > 0$ the posteriors $P_{X|Y=y}$ get predicted correctly.

**Corollary 7.** *Let the assumptions of Lemma 1 and Lemma 3 hold true and assume a global Lipschitz constant in Lemma 1. Let $p_Y$ be differentiable with $\|\nabla p_Y(y)\| \leq K$ for some $K > 0$ and all $y \in \mathbb{R}^n$. Consider a family of generative networks $(G^\varepsilon)_{\varepsilon>0}$ fulfilling*

$$\mathbb{E}_{y \sim P_Y}[W_1(P_{X|Y=y}, G^\varepsilon(y, \cdot)_\# P_Z)] \leq \varepsilon$$

*and assume that the Lipschitz constants $L^\varepsilon$ of $G^\varepsilon$ from Lemma 1 are bounded by some $L < \infty$. Then for all observations $y \in \mathbb{R}^n$ with $p_Y(y) > 0$ it holds*

$$W_1(P_{X|Y=y}, G^\varepsilon(y, \cdot)_\# P_Z) \to 0 \quad as\ \varepsilon \to 0.$$

*Proof.* We can assume that $\varepsilon \leq 1$, then the statement follows immediately from Theorem 5. $\square$

Finally, we can use Theorem 5 for error bounds of adversarival attacks on Bayesian inverse problems. Following the concurrent work of Gloeckler et al. (2023), an adversarial attack on a conditional

generative model consists in finding a perturbation $\delta$ to the observation $\tilde{y}$ so that the prediction of the conditional generative model is as far away as possible from the true posterior, i.e.,

$$\delta = \arg\max_{\|\delta\| \leq B} W_1(P_{X|Y=\tilde{y}}, G(\tilde{y}+\delta, \cdot)_{\#} P_Z).$$

Note that Gloeckler et al. (2023) use the KL divergence for their adversarial attack, but for our theoretical analysis the Wasserstein distance is more suitable.

The following corollary yields a worst case estimate on the possible attack. In the case of imperfect trained conditional generative models the attack can be very powerful depending on the strength of observation and the Lipschitz constant of the generator. If the conditional generative model is trained such that the expectation in (4) is small, then the attack can only be as powerful as the Lipschitz constant of the inverse problem allows.

**Corollary 8.** *Let the forward operator $f$ and the likelihood $p_{Y|X=x}$ in (1) fulfill the assumptions of Lemma 3. Let $\tilde{y} \in \mathbb{R}^n$ and $\delta \in \mathbb{R}^n$ with $\|\delta\| \leq B$ be an observation with $p_Y(\tilde{y}+\delta) = a > 0$. Further, assume that $y \mapsto p_Y(y)$ is differentiable with $\|\nabla p_Y(y)\| \leq K$ for $K > 0$ and all $y \in \mathbb{R}^n$. For fixed $k = \frac{a}{2K} + \|\tilde{y}\| + B$, assume that we have trained a family of generative models $G$ which fulfills $\|\nabla_y G(y,z)\| \leq L_k$ for all $z \in \mathrm{supp}(P_Z)$ and all $y \in \mathbb{R}^n$ with $\|y\| \leq k$ for some $L_k > 0$ such that*

$$\mathbb{E}_{y \sim P_Y}[W_1(P_{X|Y=y}, G(y, \cdot)_{\#} P_Z)] \leq \varepsilon$$

*for some $0 < \varepsilon \leq 1$. Then we have the following control on the strength of the adversarial attack*

$$W_1(P_{X|Y=\tilde{y}}, G(\tilde{y}+\delta, \cdot)_{\#} P_Z) \leq C_{\|\tilde{y}\|+B} B + (L_k + C_k)\frac{\varepsilon^{\frac{1}{n+1}} a}{2K} + \frac{2\varepsilon^{\frac{1}{n+1}}}{S_n(\frac{a}{2K})^n a}.$$

*Proof.* Since it holds

$$W_1(P_{X|Y=\tilde{y}}, G(\tilde{y}+\delta, \cdot)_{\#} P_Z) \leq W_1(P_{X|Y=\tilde{y}}, P_{X|Y=\tilde{y}+\delta}) + W_1(P_{X|Y=\tilde{y}+\delta}, G(\tilde{y}+\delta, \cdot)_{\#} P_Z),$$

the result follows from the application of Lemma 3 and Theorem 5. □

## 3 Conditional Generative Models

In this section, we discuss whether the main assumption, namely that the averaged Wasserstein distance $\mathbb{E}_{y \sim P_Y}[W_1(P_{X|Y=y}, G(y, \cdot)_{\#} P_Z)]$ in (4) becomes small, is reasonable for different conditional generative models. Therefore we need to relate the typical choices of training loss with the Wasserstein distance. For a short experimental verification in the case of conditional normalizing flows we refer to Appendix C.

In the following we assume that the training loss of the corresponding models become small. This can be justified by universal approximation theorems like (Teshima et al., 2020; Lyu et al., 2022) for normalizing flows or (Lu & Lu, 2020) for GANs.

### 3.1 Conditional Normalizing Flows

Conditional normalizing flows (Altekrüger & Hertrich, 2023; Andrle et al., 2021; Ardizzone et al., 2019; Winkler et al., 2019) are a family of normalizing flows parameterized by a condition, which in our case is the observation $y$. The aim is to learn a network $\mathcal{T}\colon \mathbb{R}^n \times \mathbb{R}^m \to \mathbb{R}^m$ such that $\mathcal{T}(y, \cdot)$ is a diffeomorphism and $\mathcal{T}(y, \cdot)_{\#} P_Z \approx P_{X|Y=y}$ for all $y \in \mathbb{R}^n$, where $\approx$ means that two distributions are similar in some proper distance or divergence. This can be done via minimizing the expectation on $Y$ of the *forward* KL divergence $\mathbb{E}_{y \sim P_Y}[\mathrm{KL}(P_{X|Y=y}, \mathcal{T}(y, \cdot)_{\#} P_Z)]$, which is equal, up to a constant, to

$$\mathbb{E}_{x \sim P_X, y \sim P_Y}[-\log p_Z(\mathcal{T}^{-1}(y, x)) - \log(|\det D\mathcal{T}^{-1}(y, x)|)],$$

where the inverse is meant with respect to the second component, see Hagemann et al. (2022) for more details. Training a network using the forward KL has many desirable properties like a mode-covering behaviour of $\mathcal{T}(y, \cdot)_{\#} P_Z$. Now conditional normalizing flows are trained using the KL divergence, while the theoretical bound in Section 2 relies on the metric properties of the Wasserstein-1 distance. Thus we need to show that we can ensure a small $\varepsilon$ in (4) when training the conditional normalizing flow as proposed. Following Gibbs & Su (2002, Theorem 4), we can bound the Wasserstein distance by the total variation distance, which in turn is bounded by KL via Pinsker's inequality (Pinsker, 1963), i.e.,

$$\mathbb{E}_{y \sim P_y}[W_1((P_{X|Y=y}, \mathcal{T}(y, \cdot)_{\#} P_Z)^2] \leq C \, \mathbb{E}_{y \sim P_Y}[\mathrm{TV}((P_{X|Y=y} - \mathcal{T}(y, \cdot)_{\#} P_Z)^2]$$
$$\leq \frac{C}{\sqrt{2}} \mathbb{E}_{y \sim P_Y}[\mathrm{KL}((P_{X|Y=y}, \mathcal{T}(y, \cdot)_{\#} P_Z)],$$

where $C$ is a constant depending on the support of the probability measures. However, by definition $\mathrm{supp}(\mathcal{T}(y, \cdot)_{\#} P_Z) = \mathbb{R}^m$. By Altekrüger et al. (2023, Lemma 4) the density $p_{\mathcal{T}(y, \cdot)_{\#} P_Z}$ decays exponentially. Therefore, we expect in practice that the Wasserstein distance becomes small if the KL vanishes even though (Gibbs & Su, 2002, Theorem 4) is not applicable.

### 3.2 Conditional Wasserstein GANs

In Wasserstein GANs (Arjovsky et al., 2017), a generative adversarial network approach is taken in order to sample from a target distribution. For this, the dual formulation (2) is used in order to calculate the Wasserstein distance between measures $P_X$ and $P_Y$. Then the 1-Lipschitz function is reinterpreted as a discriminator in the GAN framework (Goodfellow et al., 2014). If the corresponding minimizer in the space of 1-Lipschitz functions can be found, then optimizing the adversarial Wasserstein GAN loss directly optimizes the Wasserstein distance. The classical Wasserstein GAN loss for a target measure $\mu$ and a generator $G\colon \mathbb{R}^d \to \mathbb{R}^m$ is given by

$$\min_{\theta} \max_{\mathrm{Lip}(\varphi) \leq 1} \mathbb{E}_{x \sim P_X, z \sim P_Z}[\varphi(x) - f(G(z))],$$

where $d \in \mathbb{N}$ is the dimension of the latent space.

The Wasserstein GAN framework can be extended to conditional Wasserstein GANs (Adler & Öktem, 2018; Liu et al., 2021) for solving inverse problems. For this, we aim to train generators $G\colon \mathbb{R}^n \times \mathbb{R}^d \to \mathbb{R}^m$ and average with respect to the observations

$$L(\theta) = \mathbb{E}_{y \sim P_y}\Big[\max_{\mathrm{Lip}(\varphi_y) \leq 1} \mathbb{E}_{x \sim P_{X|Y=y}, z \sim P_Z}[\varphi_y(x) - \varphi_y(G(y, z))]\Big].$$

Hence minimizing this loss (or a variant of it) directly enforces a small $\varepsilon$ in assumption (4).

### 3.3 Conditional Diffusion Models

In diffusion models, a forward SDE, which maps a data distribution to an approximate Gaussian distribution is considered (Song et al., 2021a;b). Then the theory of reverse SDEs (Anderson, 1982) allows to sample from the data distribution by learning the score $\nabla \log p_t(x)$, where $p_t(x)$ is the path density of the forward SDE. The forward SDE usually reads

$$dX_t = -\alpha X_t \mathrm{d}t + \sqrt{2\alpha}\mathrm{d}W_t,$$

while the reverse SDE is given by

$$dY_t = -\alpha Y_t \mathrm{d}t - 2 \,\nabla \log p_t(x)\mathrm{d}t + \sqrt{2\alpha}\mathrm{d}\tilde{W}_t,$$

where $\alpha \in \mathbb{R}$ describes the schedule of the SDE. However, the path density $p_t(x)$ is usually intractable, so that the score $\nabla \log p_t(x)$ is learned with a NN $s_\theta \colon [0, T] \times \mathbb{R}^m \to \mathbb{R}^m$ such that $s_\theta(t, x) \approx \nabla \log p_t(x)$ for all $t \in [0, T]$ and $x \in \mathbb{R}^m$. This can be ensured using the so-called score matching loss (Song et al., 2021b) defined by

$$\min_\theta \mathbb{E}_{t\sim U([0,T]),x\sim P_{X_t}} \left[ \|s_\theta(t, x) - \nabla \log p_t(x)\|^2 \right].$$

In order to solve inverse problems, we can consider a conditional reverse SDE

$$dY_t = -\alpha Y_t \mathrm{d}t - 2 \,\nabla \log p_t(x|y)\mathrm{d}t + \sqrt{2\alpha}\mathrm{d}\tilde{W}_t,$$

where $p_t(x|y)$ is the conditional path density given an observation $y \in \mathbb{R}^n$. Consequently, we consider conditional diffusion models, where a NN $s_\theta \colon \mathbb{R}^n \times [0, T] \times \mathbb{R}^m \to \mathbb{R}^m$ is learned to approximate $s_\theta(y, t, x) \approx \nabla \log p_t(x|y)$ for all $t \in [0, T]$, $x \in \mathbb{R}^m$ and all observations $y \in \mathbb{R}^m$. Then the score matching loss for conditional diffusion models is given in Batzolis et al. (2021, Theorem 1) as

$$L(\theta) = \mathbb{E}_{y\sim P_Y}\left[\mathbb{E}_{t\sim U([0,T]),x\sim P_{X_t|Y=y}}[\|s_\theta(y, t, x) - \nabla \log p_t(x|y)\|^2]. \tag{9}$$

Denote by $\tilde{Y}$ the solution to the approximated SDE starting at $\tilde{Y}_0 \approx P_Z$ and $\tilde{Y}^y$ the solution of the approximated SDE conditioned on an observation $y \in \mathbb{R}^n$. Then we can use the bound derived in Pidstrigach et al. (2023, Theorem 2) which gives

$$\mathbb{E}_{y\sim P_Y}[W_2(P_{X|Y=y}, P_{\tilde{Y}_T^y})] \leq \mathbb{E}_{y\sim P_Y}\left[C\,W_2\left(P_{X_T^y}, \mathcal{N}(0, \mathrm{Id})\right)\right] + TL(\theta),$$

where $C$ is a constant depending on the length of the interval $T$ and the Lipschitz constant of the conditional score $\nabla \log p_t(x|y)$. Finally, Hölders inequality yields for the Wasserstein-1 distance

$$\mathbb{E}_{y\sim P_Y}[W_1(P_{X|Y=y}, P_{\tilde{Y}_T^y})] \leq \mathbb{E}_{y\sim P_Y}\left[C\,W_2\left(P_{X_T^y}, \mathcal{N}(0, \mathrm{Id})\right)\right] + TL(\theta).$$

Hence, when training the conditional diffusion model by minimizing (9) we also ensure that (4) becomes small. For more in depth discussion with less restrictive assumptions on the score, see also (De Bortoli, 2022).

### 3.4 Conditional Variational Autoencoder

Variational Autoencoder (VAE) (Kingma & Welling, 2013) aim to approximate a distribution $P_X$ by learning a stochastic encoder $E_\phi \colon \mathbb{R}^m \to \mathbb{R}^d \times \mathbb{R}^{d,d}$ determining parameters of the normal distribution $(\mu_\phi(x), \Sigma_\phi(x))$ for $x$ sampled from $P_X$ and pushing $P_X$ to a latent distribution $P_Z$ with density $p_Z$ of dimension $d \in \mathbb{N}$. In the reverse direction, a stochastic decoder $D_\theta \colon \mathbb{R}^d \to \mathbb{R}^m \times \mathbb{R}^{m,m}$ determines parameters of the normal distribution $(\mu_\theta(z), \Sigma_\theta(z))$ for $z \in \mathbb{R}^d$ and pushes $P_Z$ back to $P_X$. By definition, the densities of $E_\phi$ and $D_\theta$ are given by $q_\phi(z|x) = \mathcal{N}(z; \mu_\phi(x), \Sigma_\phi(x))$ and $p_\theta(x|z) = \mathcal{N}(x; \mu_\theta(z), \Sigma_\theta(z))$, respectively. These networks are trained by minimizing the so-called evidence lower bound (ELBO)

$$\mathrm{ELBO}(\theta, \phi) = -\mathbb{E}_{x \sim P_X} \left[ \mathbb{E}_{z \sim q_\phi(\cdot|x)} [\log(p_\theta(x|z) p_Z(z)) - \log(q_\phi(z|x)))] \right].$$

By Hagemann et al. (2023, Theorem 4.1), the loss $L(\theta, \phi)$ is related to KL by

$$\mathrm{KL}(P_X, D_{\theta\#} P_Z) \leq \mathrm{ELBO}(\theta, \phi).$$

We can solve inverse problems by extending VAEs to conditional VAEs (Lim et al., 2018; Sohn et al., 2015) and aim to approximate the posterior distribution $P_{X|Y=y}$ for a given observation $y \in \mathbb{R}^n$. The conditional stochastic encoder $E_\phi \colon \mathbb{R}^n \times \mathbb{R}^m \to \mathbb{R}^d \times \mathbb{R}^{d,d}$ and conditional stochastic decoder $D_\theta \colon \mathbb{R}^n \times \mathbb{R}^d \to \mathbb{R}^m \times \mathbb{R}^{m,m}$ are trained by

$$L(\theta, \phi) = \mathbb{E}_{y \sim P_Y} \left[ -\mathbb{E}_{x \sim P_{X|Y=y}} \left[ \mathbb{E}_{z \sim q_\phi(\cdot|y,x)} [\log(p_\theta(x|y,z) p_Z(z)) - \log(q_\phi(z|y,x)))] \right] \right].$$

By the same argument as above, the KL can be bounded by

$$\mathbb{E}_{y \sim P_Y} [\mathrm{KL}(P_{X|Y=y}, D_\theta(y, \cdot)_\# P_Z)] \leq L(\theta, \phi)$$

and, using similar arguments as in Section 3.1, we get the estimate

$$\mathbb{E}_{y \sim P_Y} [W_1(P_{X|Y=y}, D_\theta(y, \cdot)_\# P_Z)^2] \leq \frac{C}{\sqrt{2}} L(\theta, \phi).$$

## 4 Conclusion

We showed a pointwise stability guarantee of the Wasserstein distance between the posterior $P_{X|Y=y}$ of a Bayesian inverse problem and the learned distribution $G(y, \cdot)_\# P_Z$ of a conditional generative model $G$ under certain assumptions. In particular, the pointwise bound depends on the Lipschitz constant of the conditional generator with respect to the observation, the Lipschitz constant of the inverse problem, the training loss with respect to the Wasserstein distance and the probability of the considered observation.

The required training accuracy of the bound depends on the Wasserstein-1 distance between the target distribution and the learned distribution. However, some conditional networks as the conditional normalizing flow are not trained to minimize the Wasserstein-1 distance. Consequently, a direct dependence of the bound on the training accuracy with respect to the KL divergence would be helpful. Under very strong assumptions, the continuity in Lemma 1 has been shown for KL in Baptista et al. (2023). This could be used to derive a similar statement.

Furthermore, our bound is a worst case bound and is not always practical if the constants are large. It would be interesting to check whether tightness of the bound can be shown for some examples.

**Acknowledgement**

P.H. acknowledges funding by the German Research Foundation (DFG) within the project of the DFG-SPP 2298 "Theoretical Foundations of Deep Learning" and F.A. within project EF3-7 of Germany's Excellence Strategy – The Berlin Mathematics Research Center MATH+. The authors want to thank B. Sprungk for posing the question considered in this paper, namely if there are guarantees that conditional generative NNs work well for single observations. Moreover, many thanks to J. Hertrich for fruitful discussions and for suggesting the example in the appendix.

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

## A    Example on the Robustness of the MAP and Posterior

We like to provide an example that illustrates the stability of the posterior distribution in contrast to the MAP estimator and highlights the role of the MMSE estimator.

By the following lemma, see, e.g., (Grana et al., 2017; Hagemann et al., 2023), the posterior of a Gaussian mixture model given observations from a linear forward operator corrupted by white Gaussian noise can be computed analytically.

**Lemma 9.** *Let $X \sim \sum_{k=1}^{K} w_k \mathcal{N}(m_k, \Sigma_k) \in \mathbb{R}^m$ be a Gaussian mixture random variable. Suppose that*

$$Y = AX + \Xi,$$

*where $A : \mathbb{R}^m \to \mathbb{R}^n$ is a linear operator and $\Xi \sim N(0, \sigma^2 I_n)$. Then the posterior is also a Gaussian mixture*

$$P_{X|Y=y} \propto \sum_{k=1}^{K} \tilde{w}_k \mathcal{N}(\cdot | \tilde{m}_k, \tilde{\Sigma}_k)$$

*with*

$$\tilde{\Sigma}_k := (\tfrac{1}{\sigma^2} A^{\mathrm{T}} A + \Sigma_k^{-1})^{-1}, \qquad \tilde{m}_k := \tilde{\Sigma}_k (\tfrac{1}{b^2} A^{\mathrm{T}} y + \Sigma_k^{-1} \mu_k)$$

*and*

$$\tilde{w}_k := w_k \exp \left( \frac{1}{2} (\tilde{m}_k \tilde{\Sigma}_k^{-1} \tilde{m}_k - m_k \Sigma_k^{-1} m_k) \right).$$

Now, for some small $\varepsilon > 0$ we consider the random variable $X \in \mathbb{R}$ with simple prior distribution

$$P_X = \frac{1}{2} \mathcal{N}(-1, \varepsilon^2) + \frac{1}{2} \mathcal{N}(1, \varepsilon^2)$$

and observations from $Y = X + \Xi$ with noise $\Xi \sim \mathcal{N}(0, \sigma^2)$. The MAP estimator is given by

$$\begin{aligned}
x_{\mathrm{MAP}}(y) &\in \arg\max_x p_{X|Y=y}(x) \\
&= \arg\min_x \frac{1}{2\sigma^2} (y - x)^2 - \log \left( \frac{1}{2} (e^{-\frac{1}{2\varepsilon^2}(x-1)^2} + e^{-\frac{1}{2\varepsilon^2}(x+1)^2}) \right) \\
&= \arg\min_x \frac{1}{2\sigma^2} (y - x)^2 + \frac{1}{2\varepsilon^2} (x^2 + 1) - \log \left( \cosh \left( \frac{x}{\varepsilon^2} \right) \right).
\end{aligned}$$

The above minimization problem has a unique global minimizer for $y \neq 0$ which we computed numerically. Figure 3 (top) shows the plot of the function $x_{\mathrm{MAP}}(y)$ for $\varepsilon^2 = 0.05^2$ and different values of $\sigma$. Clearly, small perturbations of $y$ near zero lead to qualitatively completely different $x$-values, where a smaller noise level $\sigma$ lowers the distance between the values $x_{\mathrm{MAP}}(y)$ for $y > 0$ and $y < 0$. In other words, the MAP estimator is not robust with respect to perturbations of the observations near zero.

In contrast, using Lemma 9, we can compute the posterior

$$P_{X|Y=y} = \frac{1}{\tilde{w}_1 + \tilde{w}_2} (\tilde{w}_1 \mathcal{N}(\cdot | \tilde{m}_1, \tilde{\sigma}^2) + \tilde{w}_2 \mathcal{N}(\cdot | \tilde{m}_2 \tilde{\sigma}^2))$$

with

$$\tilde{\sigma}^2 = \frac{\sigma^2 \varepsilon^2}{\sigma^2 + \varepsilon^2}, \quad \tilde{m}_1 = \frac{\varepsilon^2 y + \sigma^2}{\varepsilon^2 + \sigma^2}, \quad \tilde{m}_2 = \frac{\varepsilon^2 y - \sigma^2}{\varepsilon^2 + \sigma^2},$$

$$\tilde{w}_1 = \frac{1}{2\varepsilon} \exp \left( \frac{1}{2\varepsilon^2} \left( \frac{(\varepsilon^2 y + \sigma^2)^2}{\sigma^2 (\varepsilon^2 + \sigma^2)} - 1 \right) \right), \quad \tilde{w}_2 = \frac{1}{2\varepsilon} \exp \left( \frac{1}{2\varepsilon^2} \left( \frac{(\varepsilon^2 y - \sigma^2)^2}{\sigma^2 (\varepsilon^2 + \sigma^2)} - 1 \right) \right).$$

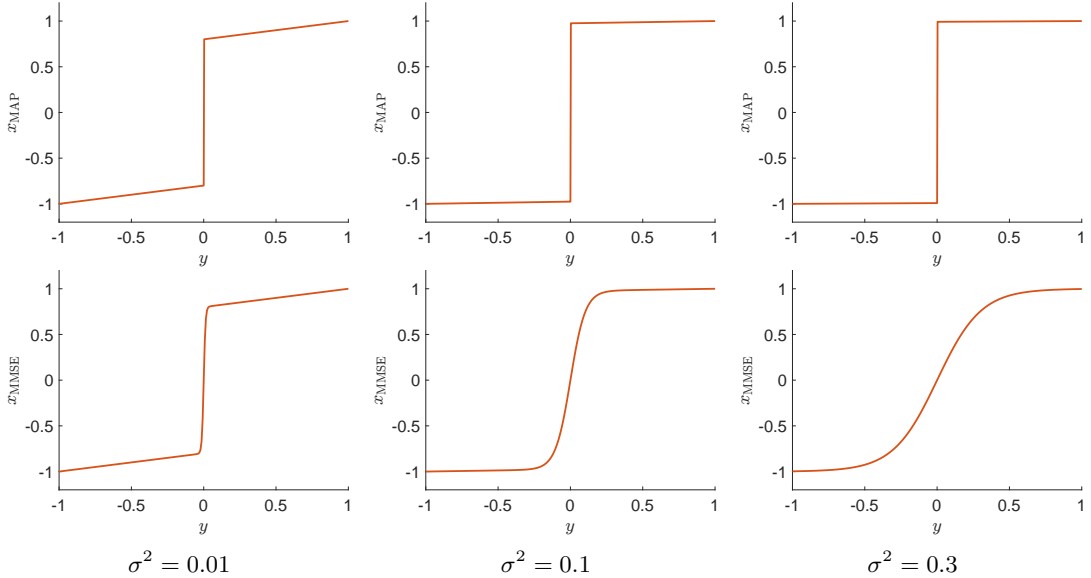

Figure 3: The MAP estimator (top) and the MMSE estimator (bottom) with respect to the observation $y$ for $\varepsilon^2 = 0.05^2$ and different noise levels $\sigma^2$.

Then the MMSE estimator is given by the expectation value of the posterior

$$
\begin{aligned}
x_{\mathrm{MMSE}}(y) &= \arg\min_{T} \mathbb{E}_{(x,y)\sim P_{(X,Y)}} \|x - T(y)\|^2 = \mathbb{E}[X|Y=y] \\
&= \int_{\mathbb{R}} x\, p_{X|Y=y}(x)\, \mathrm{d}x \\
&= \frac{1}{\tilde{w}_1 + \tilde{w}_2} (\tilde{w}_1 \tilde{m}_1 + \tilde{w}_2 \tilde{m}_2) \\
&= \frac{1}{\tilde{w}_1 + \tilde{w}_2} \frac{1}{\varepsilon(\varepsilon^2 + \sigma^2)} e^{\frac{\varepsilon^2 y^2 - \sigma^2}{2\sigma^2(\varepsilon^2 + \sigma^2)}} \left( \varepsilon^2 y \cosh(\frac{y}{\varepsilon^2 + \sigma^2}) + \sigma^2 \sinh(\frac{y}{\varepsilon^2 + \sigma^2}) \right).
\end{aligned}
$$

In Figure 3 (bottom), we see that the MMSE estimator shows a smooth transition in particular for larger noise levels, meaning that the estimator is robust against small perturbations of the observation near zero. Note that in case of a Gaussian prior $X \sim \mathcal{N}(m, \Sigma)$ in $\mathbb{R}^m$ and white Gaussian noise, the MAP and MMSE estimators coincide.

## B  Local Lipschitz continuity of the generator for a latent space with infinite support

Here we show a weakened version of Lemma 1 leading to an arbitrary small additive constant. The main difference is the weaker assumption $\|\nabla_y G(y, z)\| \leq L_r$ for all $z \in \mathbb{R}^d$ with $\|z\| \leq \tilde{r}$ and all $y \in \mathbb{R}^n$ with $\|y\| \leq r$, which is fulfilled for continuously differentiable generators. For this we use the so-called truncated normal distribution (Horrace, 2005; Tallis, 1963). Let $p_Z$ be the density of the

standard normal distribution $P_Z = \mathcal{N}(0, I_n)$, then the density of the truncated normal distribution $P_Z^{\tilde{r}}$ is given by

$$p_Z^{\tilde{r}}(z) = \begin{cases} \frac{p_Z(z)}{\int_{B_{\tilde{r}}(0)} p_Z(z)\mathrm{d}z} = \frac{p_Z(z)}{C_{\tilde{r}}}, & \text{if } \|z\| \leq \tilde{r}, \\ 0, & \text{else.} \end{cases}$$

**Lemma 10.** *Let $P_Z = \mathcal{N}(0, I_n)$ be the latent space. For any parameterized family of generative models $G$ with $\|\nabla_y G(y, z)\| \leq L_r$ for all $z \in \mathbb{R}^d$ with $\|z\| \leq \tilde{r}$ and all $y \in \mathbb{R}^n$ with $\|y\| \leq r$ for some $L_r > 0$ and some $r > 0$, it holds*

$$W_1(G(y_1, \cdot)_\# P_Z, G(y_2, \cdot)_\# P_Z) \leq L_r \|y_1 - y_2\| + M_{\tilde{r}}$$

*for all $y_1, y_2 \in \mathbb{R}^n$ with $\|y_1\|, \|y_2\| \leq r$. The additive constant $M_{\tilde{r}}$ fulfills $M_{\tilde{r}} \to 0$ for $\tilde{r} \to \infty$.*

*Proof.* Let $y_1, y_2 \in \mathbb{R}^n$ with $\|y_1\|, \|y_2\| \leq r$, then it holds

$$W_1(G(y_1, \cdot)_\# P_Z, G(y_2, \cdot)_\# P_Z) \leq W_1(G(y_1, \cdot)_\# P_Z, G(y_1, \cdot)_\# P_Z^{\tilde{r}}) + W_1(G(y_1, \cdot)_\# P_Z^{\tilde{r}}, G(y_2, \cdot)_\# P_Z^{\tilde{r}}) \\ + W_1(G(y_2, \cdot)_\# P_Z^{\tilde{r}}, G(y_2, \cdot)_\# P_Z).$$

By the assumption on the generator $G$, Lemma 1 yields

$$W_1(G(y_1, \cdot)_\# P_Z^{\tilde{r}}, G(y_2, \cdot)_\# P_Z^{\tilde{r}}) \leq L_r \|y_1 - y_2\|.$$

Consequently, it suffices to show that for $y \in \mathbb{R}^n$ with $\|y\| \leq r$ the term $W_1(G(y \cdot)_\# P_Z, G(y, \cdot)_\# P_Z^{\tilde{r}})$ vanishes for $\tilde{r} \to \infty$. By definition, it holds

$$W_1(G(y, \cdot)_\# P_Z, G(y, \cdot)_\# P_Z^{\tilde{r}}) = \max_{\mathrm{Lip}(\varphi) \leq 1} \int_{\mathbb{R}^d} \varphi(G(y, z))\mathrm{d}P_Z(z) - \int_{\mathbb{R}^d} \varphi(G(y, z))\mathrm{d}P_Z^{\tilde{r}}(z)$$

$$= \max_{\mathrm{Lip}(\varphi) \leq 1} \int_{\mathbb{R}^d \setminus B_{\tilde{r}}(0)} \varphi(G(y, z))\mathrm{d}P_Z(z) + \int_{B_{\tilde{r}}(0)} \varphi(G(y, z))\mathrm{d}P_Z(z)$$

$$- \int_{B_{\tilde{r}}(0)} \varphi(G(y, z))\mathrm{d}P_Z^{\tilde{r}}(z)$$

$$= \max_{\mathrm{Lip}(\varphi) \leq 1} \int_{\mathbb{R}^d \setminus B_{\tilde{r}}(0)} \varphi(G(y, z))p_Z(z)\mathrm{d}z$$

$$+ \int_{B_{\tilde{r}}(0)} \varphi(G(y, z))p_Z(z)(1 - \frac{1}{C_{\tilde{r}}})\mathrm{d}z.$$

The first term vanishes exponentially in $\tilde{r}$ by the density $p_Z$, and for the second term note that $C_{\tilde{r}} \to 1$ for $\tilde{r} \to \infty$. $\square$

## C   Experimental demonstration of assumption (4)

Here we experimentally demonstrate that the expectation in (4) gets small when training a conditional generative network. For this, we consider a conditional normalizing flow as an example, consisting of three and ten Glow coupling blocks[1] with fully connected subnets with one hidden

---

[1]available at `https://github.com/VLL-HD/FrEIA`

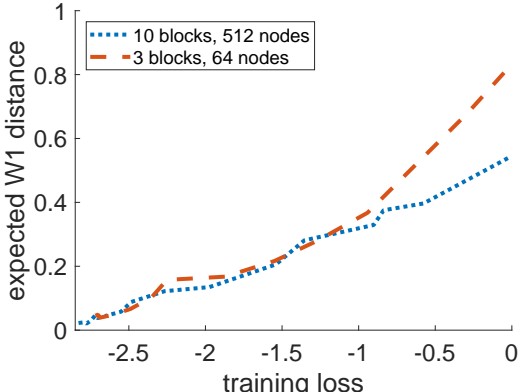

Figure 4: Expectation of the Wasserstein distance between posterior and pushforward of the generator with respect to the training loss of the conditional normalizing flow.

layer and 64 and 512 nodes, respectively. As in Appendix A, we choose a two-dimensional Gaussian mixture model with six modes as prior distribution and additive Gaussian noise with standard deviation 0.5 for the noise model. The forward operator is chosen to be the identity. Then we train the conditional normalizing flow for 100000 optimizer steps using Adam Kingma & Ba (2015) with a learning rate of $1e-4$ and a batch size of 1024. Note that we can analytically compute the posterior distribution by Lemma 9.

In Figure 4 we visualize the expectation in (4) with respect to the training loss of the conditional normalizing flow, which is, up to a constant, equal to the KL, see Section 3.1. Obviously, the expectation (4) gets small when minimizing the training loss of the conditional normalizing flow. We computed the Wasserstein distance using POT [2] and draw 20000 samples from each distribution. Moreover, we discretized the expectation by drawing 30 observations.

---

[2]available at `https://github.com/PythonOT/POT`

