# OpenReview forum: "Conditional Generative Models are Provably Robust: Pointwise Guarantees for Bayesian Inverse Problems"
_TMLR — Accepted by TMLR_

### Review · Reviewer_SurZ · 2023-05-05

**Summary Of Contributions:**

The article characterises the robustness of broad range of conditional generative models in the task of Bayesian inverse problems. The paper provides a general theorem about pointwise robustness for a fairly general family of generative models, in form of Theorem 5 that provides an upper bound for Wasserstein distance between the true solution and the estimate in terms of smoothness of the inverse problem and the generator, as well as a threshold (again expressed in terms of the Wasserstein distance) for the quality of the generator. This general result is then related to several practical families of conditional generative models, explaining how normalizing flows, GANs, diffusion models, and VAEs can be analysed in this framework.

**Audience:**

Yes

**Broader Impact Concerns:**

No need for discussing broader impact in this paper; it is obvious that improved theoretical understanding of the robustness of a computational approach is useful for improved transparency and quality of future research and bears no risks.

**Claims And Evidence:**

Yes

**Requested Changes:**

I request the authors to:
1. Carefully check the Introduction. Remove/move the first paragraph and try to streamline the story in the 2nd and 3rd paragraph. The core contribution of the paper is in analysis of how well some methods work for the Bayesian inverse problem and hence the core audience will be people already familiar with the reasons we solve that problem. Write for that audience.

2. Bring the motivational example in Appendix A to the main paper. This can be done in many ways (some suggestions provided above), but I think something has to be done to make the paper work. The current approach where you refer to that example several times to motivate the work but still leave the whole description in Appendix is problematic in too many ways.

**Strengths And Weaknesses:**

The paper provides a clear theoretical result that characterises the robustness of a broad class of approaches, effectively proving that some computationally appealing approaches used in general machine learning literature are applicable for solving Bayesian inverse problems. The result is nontrivial and relates intuitive concepts (Lipschitz constants and the training objective of the generator) to the upper bound on approximation error, and is likely to be useful both for practitioners and for further theoretical understanding of the the problem. The paper is in general well written and easy to understand for a theoretical paper and related work is covered well, and the claims are well justified by detailed theorems and proofs. There is no need for the paper to have empirical results of any kind.

The other main strength is detailed presentation of how the general result looks like in context of specific families of conditional generative models, covering effectively all of the main families currently used in the field. I find it highly useful that e.g. GANs and diffusion models are interpreted within the same framework, since this improves our understanding of their relationships more broadly, not just in the context of the specific setting (here Bayesian inverse problems). I commend the authors for taking the otherwise fairly general theorem to the level of practical application via these examples.

The only minor weaknesses concern the structure and presentation, detailed below:
1. The structure and flow of Introduction is confusing. The first paragraph that talks about vulnerability of NNs and adversarial attacks is completely out of place here. Does it even belong in this paper? Even if the answer is yes, it definitely should not be the first paragraph, but you should directly start from the task of Bayesian inverse problems. Somewhat related to this, I do not see need for relating the work that much to 'end-to-end solutions' -- you are anyway doing the work in context of Bayesian inverse problems and can just start from stating that you are providing the Bayesian solution with the obvious advantages (as pointed out by numerous references) it has, and then proceed to explain how conditional generative models offer nice opportunities for the task but their theoretical properties are not yet known well enough.
2. You refer in the main paper couple of times to the motivational example in Appendix. This is quite difficult for the reader since it requires jumping back and forth. If you think that the example is needed for clarifying the motivation, to the extent that you want to refer to it already in the second paragraph of the paper, then it should be in the main paper. My practical recommendation is to create a version that combines essential content of Figures 1 and 2 (for instance, one column of Fig 1 and 1-2 examples from Fig 2) and put it on the second page of the main paper. Then you can explicitly refer to that image and explain what we see there in Introduction. You can still leave the details of the model (Lemma 8 etc) in the Appendix as long as you just say in words in the main paper how this example corresponds to noisy linear forward operation for a mixture distribution. Alternatively, you could move the whole Appendix to become new Section 2 right after the Introduction, as an example that provides the reader the context needed for the theoretical development.

---

> ### Author Response · Authors · 2023-06-08
>
> Thank you very much for your review and the suggestions. We revised the introduction and removed the first two paragraphs in order to clearly focus on Bayesian inverse problems. Moreover, we moved Figure 2 to the second page, such that the introduction is more self-
> contained. To clarify the connection of our bound to adversarial attacks on conditional
> generative models, we established Corollary 8. Basically, under the conditions of our bound
> we can control the strength of an adversarial attack, i.e., give an upper bound on how much
> one can deviate from the posterior at a point $\tilde y$ by adding a small perturbation. We hope
> this clarifies the relation to adversarial attacks we had in mind.

---

> > ### Comment · Reviewer_SurZ · 2023-06-15
> > **Improved presentation**
> >
> > Thank you for the response and the updated paper. I think the paper now reads better and I see no reason for major changes.
> >
> > As a very minor thing, the first sentence of Introduction quite odd as the interest in quantifying uncertainty is not 'recent' but rather something statistics has been addressing for a few centuries. It's only the use of generative NNs for this task that can be considered recent.

---

> > > ### Author Response · Authors · 2023-06-15
> > >
> > > Thank you for the hint, we will update the manuscript accordingly.

---

### Review · Reviewer_MHWT · 2023-05-16

**Summary Of Contributions:**

This paper derives theoretical results on the stability of approximate posteriors for Bayesian inverse problems based on deep generative models with respect to perturbations in the observations. The results are based on the Lipschitz continuity of the generative models and its effects on the Wasserstein distance between the true posterior and the approximate one. Applications to popular deep conditional generative models are presented.


**Audience:**

Yes

**Broader Impact Concerns:**

N/A, paper of a theoretical nature

**Claims And Evidence:**

Yes

**Requested Changes:**

I acknowledge that there is already some discussion on the assumption in Eq. 4 in the paper. However, the section with the application of the theoretical results to me seems to need a more concrete demonstration that the assumptions are applicable to general models, maybe with an experimental demonstration, for example.

There are also a few minor issues: unusual quotation marks, quite a few important references are ArXiv preprints, such as Ardizzone et al. (2019), Bartzolis et al. (2021), etc., and in-text citations should use appropriate formatting, e.g., "in Author (year)", instead of "in (Author, year)".

**Strengths And Weaknesses:**

The paper is mostly well written and the theoretical results seem solid, though I haven't been able to carefully check all the proofs. The main issue to me seems to be the assumption of a well trained model satisfying the inequality in Eq. 4. I'm not sure how feasible it is to verify this condition in general for practical models. The applications section does not discuss, for example, how a small
 would be achievable for models of different complexities, as measured by, e.g., the number of layers in a conditional normalizing flow. It is, instead, based on the assumption that training reaches a reasonable approximation error level, which could be hard to verify as Eq. 4 assumes an expectation over the entire data distribution.

---

> ### Author Response · Authors · 2023-06-08
>
> Thank you very much for your review. We added a discussion towards our assumption in Eq. 4. in the beginning of Section 3 and an experimental demonstration in Appendix C for the case of conditional normalizing flows. An extensive empirical study is out of reach
> and left for future research, but in Appendix C we show that training two sets of conditional normalizing flows with KL reaches a small expected Wasserstein distance. We think it is reasonable to expect conditional generative models to reach a small expected Wasserstein
> distance, given their practical success. We also fixed the fault with the quotation marks and the unusual in-text citations.

---

### Review · Reviewer_3wGv · 2023-06-01

**Summary Of Contributions:**

The authors consider the problem of learning posterior distributions in inverse problems. The authors show that assuming conditional generative models can minimize the Wasserstein distance to the true posterior distribution in expectation over the measurements, they can guarantee that the conditional generative model is close to the posterior distribution pointwise in the observations.

**Audience:**

Yes

**Claims And Evidence:**

Yes

**Requested Changes:**

- Please state assumptions outside the Lemma statement, so that it's easier to identify what the assumptions are.

- There needs to be more concentrated effort towards making the Theorem statements readable. It is very difficult to judge the quality of the bounds in their present form.

**Strengths And Weaknesses:**

Strengths:
- I think the introduction is very well written, and the considered problem is interesting.

- In section 3, the authors show that they can transfer their guarantees, which in Theorem 5 are for conditional generative models minimizing the 1-Wasserstein distance, to different generative models that minimize different statistical distances.

Weaknesses:
- The notation / writing is somewhat confusing:
    - On the top of page 5, what is $G^{\epsilon}$?
    - Theorem 5 is very difficult to parse. The definition of $\epsilon$ makes it hard to identify how small it is, and then the main bound in Eqn 5 is also difficult to parse.
    - The authors keep using the example of additive Gaussian noise in the observations $y$, but the Lemmas, Theorems, and Assumptions are for $ \| y \| \le r$, which wouldn't be satisfied with Gaussian noise.

- The Lipschitz constant $L_r$ in Lemma 2 and the constant $C_r$ in Lemma 3 are treated as universal constants. But they may have a dimension dependence -- for example, if the generative model is a feedforward ReLU network, as $L_r$ is defined using $\| \nabla_y G(y, \cdot) \| \le L_r$, this gradient can very well scale as $ c^{poly(m,n)}$, where $c$ is the max weight in $G$, and $n, m$ are the dimensions of $y, x$ (there will also be an exponential dependence on the number of layers in the generative model). This has not been considered in any of the results.

- The strength of Theorem 5 is difficult to judge. Eqn 4 assumes that on average over $y$, the conditional generative model is close to the true posterior in Wasserstein distance (there are no algorithmic guarantees here, this is assumed to be true). Then, the authors show that for $y$ small enough (within a radius $k$, where $k$ depends on $a$, the marginal likelihood of $p_Y$ at the observed $y$), the Wasserstein distance between the posterior on this single sample $y$ is close to the true posterior distribution. It's dificult to judge how small these radiuses are (since $a$ can be exponential small for Gaussian distributions), how large the Lipshitz constants are, how small $\epsilon$ is above Eqn 5 and as a result, how good a bound the RHS of Eqn 5 is.

- I have some concerns about how well posed the problem is. The authors assume that in expectation over $y$, Eqn 4 gives $\epsilon$ Wasserstein distance between the trained model and the true posterior distribution. Then they claim that the posterior distributions are close pointwise in $y$ -- this doesn't make sense since a single $y$ is a zero-measure set. Are the results almost surely over $y$? Are they with high probability over $y$? I think the authors need to be more precise about what exactly their probabilistic claims are.

- Eqn 4 is assumed to be true for the population average over $y$, not just for a training set of $y$.

- The analysis techniques are limited. The authors assume that the generative models have good Lipshitz constants, and don't discuss what models would have such Lipshitz constants. The results also end up being multiple triangle inequalities with Lipshitz inequalities employed strategically : to show that the conditional distribution at $y$ is close to the true distribution, the authors use the triangle inequality + symmetrization to make this bound between $y_1$ and $y_2$ that are bounded, then apply the Lipshitz inequality of $G$ between $y_1$ and $y_2$. This assumes that $y_1$ and $y_2$ lie in a small enough ball, and that the Lipshitz constant is small enough to make the error bound reasonable.

---

> ### Author Response · Authors · 2023-06-08
>
> Thank you very much for your review. We answer your comments in the order they were posed.
> - We introduced the notation of $G^\varepsilon$. Moreover, we added explanations on the assumptions in our statements for a better readability.
> Please note that in our statements we consider a fixed observation $\tilde y$. Thus once fixed, it is bounded, even for additive Gaussian noise.
> - Of course, the Lipschitz constant depends on the architecture of the generator. As argued in the discussion before Remark 4, we can expect for "good" models that the Lipschitz constant of the generator behaves like the one of the inverse problem.
> Moreover, there are approaches ensuring a small Lipschitz constant of the generator.
> We added an explanation on this.
> - We added an empirical illustration for the assumption in Eq. 4 in Appendix C. Please note that we do not assume that the norm of the observation $\tilde y$ is small, for which we provide the closeness of the posterior. Instead, we fix an observation $\tilde y$ and assume that the generator fulfills the required Lipschitz property on a sufficient large set. To make this more clear, we reformulated Theorem 5 by fixing a $k$ for which we get the Lipschitz constant.
> Theorem 5 relates the pointwise distance to intuitive parameters of both the generative model and the inverse problem. In practice it is non-trivial to determine those constants. However, two things can be said: for any inverse problem we can find a training loss $\varepsilon$ so that we can predict the posterior of an in-distribution $\tilde{y}$ up to an arbitrary error. Furthermore, our bound improves interpretability as it relates the distance to well-known concepts in a quantitative way. As we are the first to consider this problem, there are plenty fruitful follow up directions both on the theoretical and practical side, as the reviewer suggested.
> - In general, we do not have any information about pointwise estimates when minimizing the expectation over $P_Y$; now, in Theorem 5 we give a pointwise (and not probabilistic) estimate under certain assumptions.
> Note that in our case the appearing densities are continuous such that the posterior density is well-defined for all observations $y \in \mathrm{supp} P_Y$.
> This is also implied by the Lemma 3 as the posterior is locally Lipschitz in the [Sprungk, 2020] setting. We clarified this in the text.
> - Please note that empirical measures approximate the compactly supported measures in rate $n^{-1/d}$ with respect to $W_1$, where $n$ is the number of samples and $d$ the dimension; see, e.g. [Weed \& Bach, 2019]. Nevertheless, this would overly complicate the theorem and is not the scope of our work. We added a comment on this.
> - There is several literature enforcing a small Lipschitz constant of generative models.
> We added a short discussion towards the Lipschitz constants and corresponding literature after Lemma 1.
> Moreover, the proof of the main theorem is non-trivial and has a nice geometric interpretation. We added an illustration in Figure 2.
>
> Jonathan Weed and Francis Bach. Sharp asymptotic and finite-sample rates of convergence of empirical measures in Wasserstein distance. Bernoulli, 25(4A):2620 – 2648, 2019
>
> Björn Sprungk. On the local Lipschitz stability of Bayesian inverse problems. Inverse Problems, 36
> (5), 2020.

---

### Author Response · Authors · 2023-06-08
**General answer**

We would like to thank all reviewers and the action editor for the thorough evaluation of the paper. We answer each reviewer separately. Moreover, we uploaded a revised version of the manuscript, where changes are highlighted in blue.

---

### Decision · Action_Editors · 2023-07-17

**Recommendation:** Accept as is

**Comment:**

While the reviewers are all in agreement leaning towards accepting this manuscript, they have not advocated for any certifications. I personally would recommend that the authors consider sharing their code in future submission.

**Audience:**

Bayesian inverse problems cover a wide range of applications, ranging from medical imaging to other forms of image and sensor processing. The Bayesian approach introduces a much needed focus on uncertainty in such applications. This paper should find an audience among TMLR readers working on such problems with a consideration for quantifying uncertainty.

**Claims And Evidence:**

The authors present a set of theoretical results quantifying the robustness of conditional generative models as applied to Bayesian inverse problems. All three reviewers have studied the authors' results and have found them to be theoretically interesting and practically useful. The authors have reviewed and revised their manuscript to take into account the clarifications requested by the reviewers; the revised manuscript offers increased clarity into the authors' findings.